# Multiple hydrogen-bonding induced nonconventional red fluorescence emission in hydrogels

Jiayu Wu[1,2], Yuhuan Wang[3], Pan Jiang [1] ✉, Xiaolong Wang [1] ✉, Xin Jia[2] & Feng Zhou [1]

The development of unconventional long-wavelength fluorescent polymer hydrogels without using polycyclic aromatic hydrocarbons or extended $\pi$-conjugation is a fundamental challenge in luminescent materials owing to a lack of understanding regarding the spatial interactions induced inherent clustering-triggered emission under water-rich conditions. Inspired by the color change of protein astaxanthin as a result of heat-induced denaturation, we propose a thermodynamically driven strategy to develop red fluorescence (~610 nm) by boiling multiple hydrogen-bonded poly(*N*-acryloylsemicarbazide) hydrogels in a water bath. We reveal that thermodynamically driven conformational changes of polymer chains from isolated hydrogen bonding donor-acceptor structures to through-space interaction structures induce intrinsic fluorescence shifts from blue to red during clustering-triggered emission. The proposed multiple hydrogen-bonding supramolecular hydrogel shows good fluorescence stability, mechanical robustness, and 3D printability for customizable shaping. We provide a viable method to prepare nonconventional long-wavelength fluorescent hydrogels towards soft fluorescent devices without initially introducing any fluorescent components.

From primitive long-afterglow phosphorescent minerals found in nature to advanced synthetic fluorescent materials[1–4], light has been utilized in various forms to promote the advancement of human society. Among such materials, fluorescent polymer hydrogels (FPHs) composed of three-dimensional cross-linked hydrophilic polymer networks have gained popularity in various fields, such as fluorescent sensing, probes, information encryption, and soft robotics[5–8]. Moreover, the soft and wet nature and favorable biological affinity of FPHs make them suitable for various applications[5,9–11]. Traditional FPHs are prepared by mixing or grafting organic fluorophores[12,13], rare earth complexes[14,15], and luminescent nanoparticles[16,17]. However, these desirable chromophores commonly require sophisticated design processes and organic

synthesis and may suffer from limited biocompatibility. Although fluorescent protein hydrogels consisting of biologically similar components exhibit natural fluorescence stability and biocompatibility[18,19], their weak mechanical properties and complex preparation processes limit their applicability.

In recent years, clustering-triggered emission (CTE) in the absence of traditional large $\pi$-conjugated structures has been introduced for the construction of FPH systems[20–24]. The mutual aggregation of $\pi$ or $n$ electron groups (such as hydroxyl, ester, carbonyl, and amide groups) has resulted in the formation of through-space interaction (TSI)[25–27], which could induce intermolecular or intramolecular electron delocalization to generate intrinsic cluster-generated chromophores.

[1]State Key Laboratory of Solid Lubrication, Lanzhou Institute of Chemical Physics, Chinese Academy of Sciences, Lanzhou 730000, China. [2]School of Chemistry and Chemical Engineering/State Key Laboratory Incubation Base for Green Processing of Chemical Engineering, Shihezi University, Shihezi 832003, China. [3]School of Chemical Sciences, University of Chinese Academy of Sciences, Beijing 100049, China. ✉e-mail: pan.jiang@ijm.fr; wangxl@licp.cas.cn

Although increasing efforts have aimed to extend luminescence regions to red and even near-infrared (NIR) regions towards high penetration capacity and resolution[28,29], the emission bands of current FPHs are still in the blue or yellow light regions[22,30–32]. In addition, a large amount of water in the hydrogel matrix has been shown to hydrate hydrophilic polymer chains and electron-rich groups, thus hindering CTE. For example, a polyacrylamide (PAAm) hydrogel network in the dry state was found to emit strong blue fluorescence, but this emission disappeared upon swelling of the material[21,33] because water molecules induced cluster disaggregation and disrupted the effect of electron delocalization. Regulating of hydrogen bonds between water molecules and polymer chains is crucial in intrinsically fluorescent hydrogel systems. However, these systems only exhibit $n-\pi^*$ transitions, rather than $\pi-\pi^*$ transitions, to achieve short-wavelength luminescence. Therefore, the utilization of CTE to extend the intrinsic luminescence to long-wavelength regions (e.g., red and even NIR regions) in water-saturated hydrogels is still a significant challenge.

In nature, many crustaceans, such as crabs and crayfish, contain a large amount of astaxanthin, and the color of the carapace of these animals changes from cyan to red during high-temperature cooking due to the thermal denaturation of proteins and the detachment of astaxanthin (Fig. 1a)[34,35]. Inspired by this, we assume that electron delocalization could be enhanced by removing the water molecules that hydrated luminescent clusters in hydrogels through thermodynamic regulation. However, removing water molecules that interact with luminescent clusters while retaining other water molecules in the polymer matrix seems paradoxical.

Here, we employ a supramolecular poly(N-acryloylsemicarbazide) (PNASC) hydrogel with strongly hydrophobic urea groups to form hydrophobic clusters within a hydrophilic network, which therefore displays traditional blue fluorescence upon excitation at 365 nm. An emergent phenomenon is that the PNASC hydrogel exhibits brilliant red fluorescence without structural collapse after processing the material in a hot water bath (above 85 °C). These results indicate that heat-induced rearrangement and aggregation promote the formation of H-bonding donor–acceptor (D–A) structures between polymer chains by strong multiple H-bonding interactions. Besides, a small fraction of heteroatoms (O and N) containing lone-pair electrons that do not form strong H-bonds may also participate in $n-n$, $n-\pi$ interactions of TSI. Thus, electron-rich groups can lead to strong TSI and conformational rigidity, avoiding the disturbance of water in hydrogel matrix. Overall, the proposed nonconventional red fluorescent hydrogel is expected to be widely used in various applications, such as bioimaging, detection, and soft robotics.

## Results

### Preparation of fluorescent hydrogels

Fluorescent hydrogels were prepared via an easy method. N-Acryloylsemicarbazide (NASC)[36] as a monomer and lithium phenyl-2,4,6-trimethylbenzoylphosphinate (LAP) as an initiator were dissolved in a mixed solvent (deionized (DI) water and dimethyl sulfoxide (DMSO)). The precursor hydrogel was first cured by 405 nm light and then soaked in DI water at 25 °C to achieve H-bond reconstruction via phase conversion. The multiple H-bonding hydrogels exhibited blue fluorescence upon 365 nm excitation as common. However, upon phase conversion in a hot water bath ($T = 85$-100 °C), the hydrogel at equilibrium unexpectedly exhibited red fluorescence upon 365 nm excitation (Supplementary Fig. 1). As shown in Fig. 1b, the fluorescence region of the crayfish-shaped hydrogel shifted from blue to

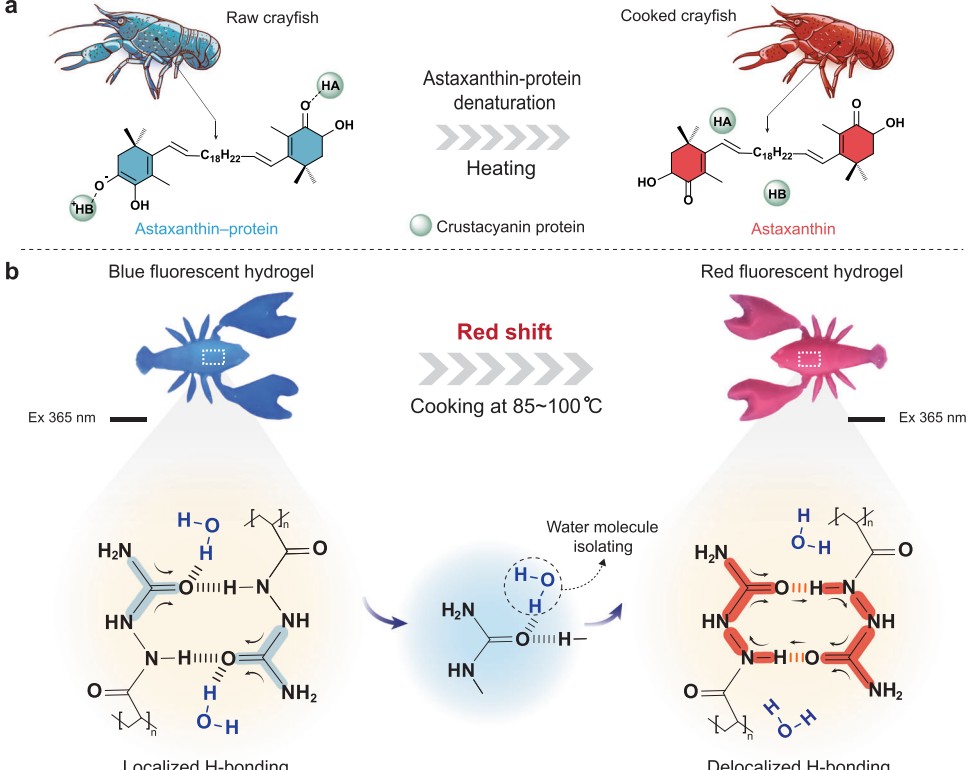

**Fig. 1 | Schematic illustration of delocalized H-bonding induced nonconventional red fluorescent in hydrogels inspired by astaxanthin-protein denaturation. a** Schematic illustration of the astaxanthin-protein denaturation process. The blue and red colors that fill in the molecule structure indicate the colors of the corresponding compounds before and after heating, respectively. **b** Delocalized processes of H-bonding in hydrogels and its corresponding crayfish-shaped hydrogel image. The scale bar is 1 cm. The blue and red shadings on the molecular structures represent the emission fluorescent colors and electron delocalization extent of corresponding polymer configurations, respectively.

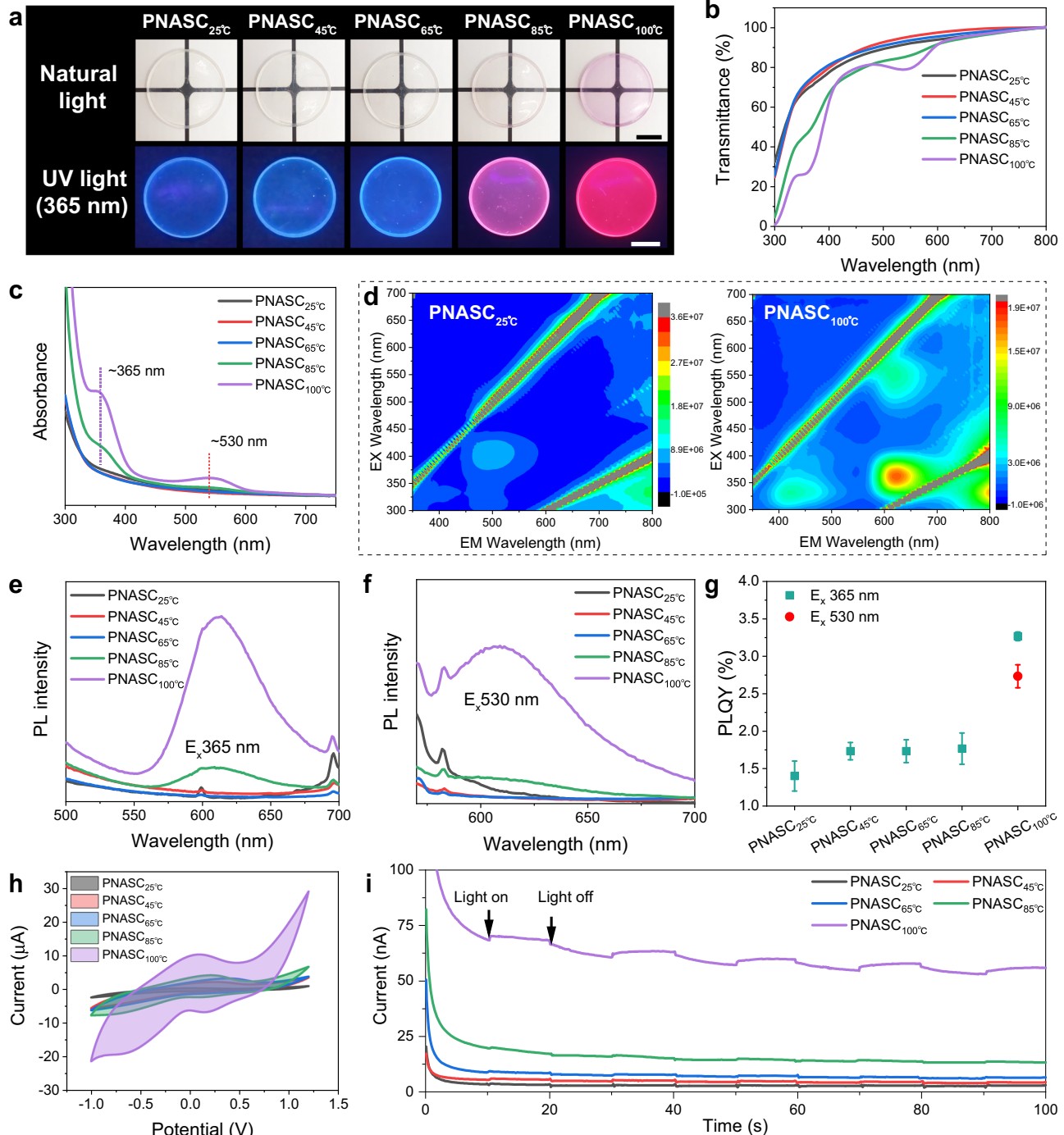

**Fig. 2 | Luminescent performance of hydrogels. a** Digital photos of PNASC hydrogel heated at different temperatures under natural light and UV light (365 nm), the scale bar is 1 cm. **b** Transmittance and **c** absorbance curves of PNASC hydrogels heated at different temperatures. **d** Excitation-emission contour plots images of PNASC$_{25°C}$ and PNASC$_{100°C}$. **e, f** Emission spectra of PNASC hydrogel excited at 365 nm and 530 nm, respectively. **g** PLQY (%) of PNASC powders excited with 365 nm and 530 nm light. **h** Cyclic-voltammetric (CV) curves of PNASC hydrogels. **i** Current-time curves of PNASC hydrogels with periodicity light on and off. Excited at 365 nm. Error bars represent the standard deviation from three replicates. Data in (**g**) are presented as mean values ± SD.

dazzling red upon 365 nm excitation after heating. This interesting phenomenon occurred because the thermodynamically driven conformational change of the polymer chains induced the formation of a cluster domain via the formation of strong multiple H-bonds between hydrophobic urea groups. Therefore, the clustered structures protected against the influence of water molecules to induce greater electron delocalization through the formation of strong multiple H-bonds, leading to red fluorescence in a water-saturated hydrogel matrix.

## Luminescence of the hydrogels

The heating temperature is a crucial factor for controlling the red fluorescence properties of hydrogels. In this case, the hydrogels treated at different temperatures for 48 h were named PNASC$_{25°C}$, PNASC$_{45°C}$, PNASC$_{65°C}$, PNASC$_{85°C}$, and PNASC$_{100°C}$ (NASC monomer concentration of 25 wt%). As shown in Fig. 2a, the hydrogels emitted blue luminescence, which became slightly brighter with increasing heating temperature. Red fluorescence was observed when the heating temperature exceeded approximately 85 °C. When the heating temperature reached

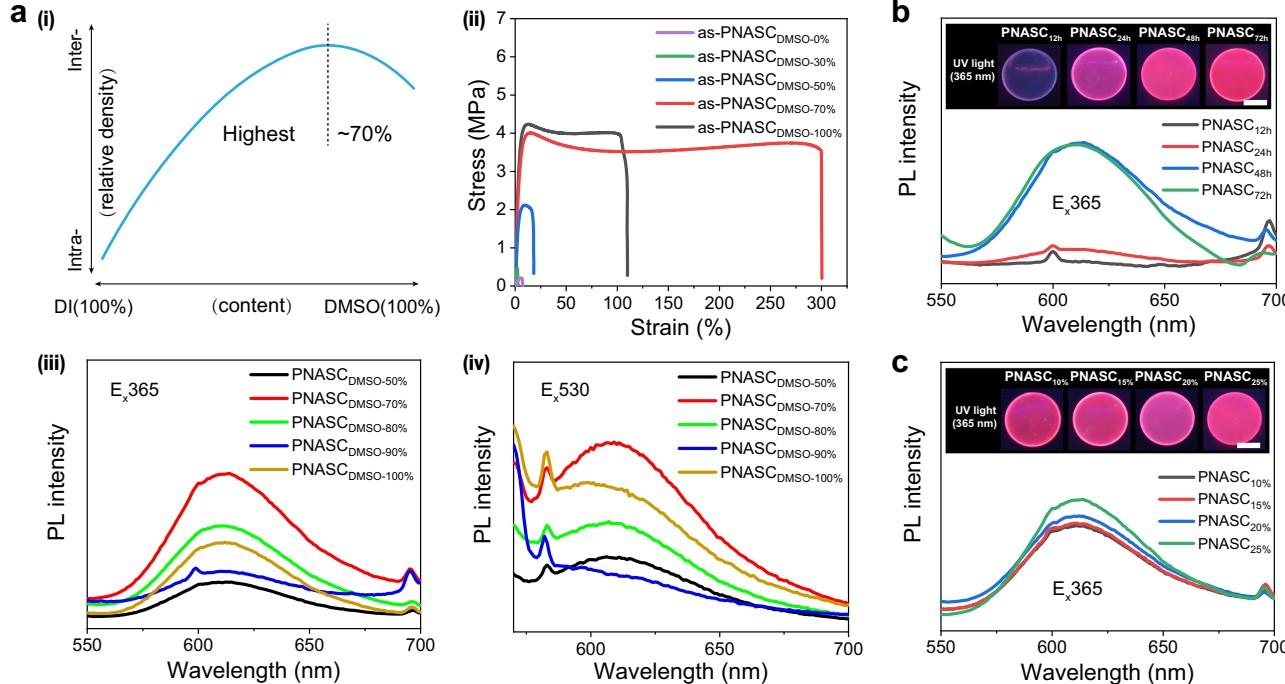

**Fig. 3 | Structure-fluorescence correlation. a** Effect of different solvent ratios on fluorescent hydrogels. (i), Schematic diagram of the relationship between different solvent ratios and the relative content of inter-/intra-chain H-bonds in polymers. (ii), Tensile-strain curves of as-PNASC$_{DMSO-0\%}$, as-PNASC$_{DMSO-30\%}$, as-PNASC$_{DMSO-50\%}$, as-PNASC$_{DMSO-70\%}$, and as-PNASC$_{DMSO-100\%}$. Emission spectra images of PNASC$_{DMSO-50\%}$, PNASC$_{DMSO-70\%}$, PNASC$_{DMSO-80\%}$, PNASC$_{DMSO-90\%}$, and PNASC$_{DMSO-100\%}$ of excited at (iii) 365 nm and (iv) 530 nm. **b** Effect of different heating time on fluorescent hydrogels. Emission spectra images of PNASC$_{12h}$, PNASC$_{24h}$, PNASC$_{48h}$, and PNASC$_{72h}$. The inset is digital photos of hydrogels under UV light (365 nm). The scale bar is 1 cm. **c** Effect of different monomer contents on fluorescent hydrogels. Emission spectra images of PNASC$_{10\%}$, PNASC$_{15\%}$, PNASC$_{20\%}$, and PNASC$_{25\%}$ of excited at 365 nm. The inset is a digital photo of hydrogels under UV light (365 nm). The scale bar is 1 cm.

100 °C, the hydrogel gave rise to obvious absorption peaks at ~365 nm and ~550 nm, corresponding to the slight red color observed under natural light (Fig. 2b, c). The results of Fourier transform infrared (FT-IR) spectroscopy and nuclear magnetic resonance (NMR) spectroscopy confirmed that the cooking process affected only the H-bonding interactions of the polymer chains instead of the chemical components (Supplementary Figs. 2, 3, and 4). Even after the samples were heated at 200 °C, the fluorescence of the PNASC hydrogel powder remained stable, indicating its excellent thermal stability (Supplementary Fig. 5). Compared with traditional fluorescent hydrogels with ultralow water contents for CTE, the proposed steady-state fluorescent hydrogel possessed a moderate water content of up to ~50% (Supplementary Fig. 6). Accordingly, the developed red fluorescent hydrogel also exhibited good flexibility compared with traditional dry fluorescent polymers or suspensions and could be folded and bent without reducing its fluorescence (Supplementary Figs. 7, 8, Supplementary Movie 1).

In contrast to the excitation-emission contour plot of PNASC$_{25°C}$, which exhibited an emission center at ~490 nm upon excitation at ~390 nm, the excitation-emission contour plot of PNASC$_{100°C}$ exhibited two emergent emission centers (~608 nm upon ~365 nm excitation and ~610 nm upon ~530 nm excitation) (Fig. 2d). As the heating temperature increased from 25 °C to 100 °C, the photoluminescence quantum yield (PLQY) of the fluorescent hydrogels increased from $1.4 \pm 0.2\%$ to $3.3 \pm 0.1\%$ upon excitation at 365 nm and from 0% to $2.7 \pm 0.2\%$ upon excitation at 530 nm (Fig. 2e–g). Besides, the fluorescence lifetime also increased from $\tau = 3.8$ ns for the PNASC$_{25°C}$ hydrogel to $\tau = 17.1$ ns for the PNASC$_{100°C}$ hydrogel (Supplementary Fig. 9). In addition, as shown in Fig. 2h and Supplementary Fig. 10, the current–voltage (CV) curve demonstrated that the redox performance of the fluorescent hydrogels gradually increased with increasing heating temperature. These results indicated that the enhanced electron delocalization ability endowed the fluorescent hydrogel with better electrical conductivity, consistent with the trend of the fluorescence intensity. In addition, the enhanced conductivity of the fluorescent hydrogels could be regulated by periodic light stimulation (Fig. 2i).

## Structure-fluorescence correlation

To understand the generation process of nonconventional red fluorescence in the proposed hydrogel system, we investigated in detail the inherent relationship between the interactions of the polymer chains and the intrinsic fluorescence. The hydrogels were first prepared from precursors at different amounts of DMSO (good solvent for NASC) and DI water (poor solvent for NASC) from 0%-100% DMSO, and the resulting equilibrium hydrogels were named as-PNASC$_{DMSO-0\%}$, as-PNASC$_{DMSO-30\%}$, as-PNASC$_{DMSO-50\%}$, as-PNASC$_{DMSO-70\%}$, and as-PNASC$_{DMSO-100\%}$. As shown in Supplementary Fig. 11 and Supplementary Fig. 12, the solvent ratio dramatically affected the conformation of the PNASC polymer chains. It was found that as-PNASC$_{DMSO-70\%}$ had the highest transparency and lowest water content, indicating that it had more homogeneous and densified polymer networks than the other materials (Supplementary Fig. 13-15). Therefore, we believe that as-PNASC$_{DMSO-70\%}$ possessed the highest density of interchain hydrogen bonds (Fig. 3a(i)). The highest glass transition temperature ($T_g$) of as-PNASC$_{DMSO-70\%}$ was also confirmed by differential scanning calorimetry (DSC) and was attributed to the broad interchain H-bond density of this sample (Supplementary Fig. 16). Moreover, as-PNASC$_{DMSO-70\%}$, which possessed abundant interchain H-bonds, exhibited high strength and elongation due to the energy dissipation of the hydrogen bonds between the polymer chains (Fig. 3a(ii), Supplementary Fig. 17). Interestingly, as shown in Supplementary Fig. 18, there was a strong correlation between the fluorescence performance and the polymer conformation of the hydrogel after heating for 48 hours (named PNASC$_{DMSO-50\%}$, PNASC$_{DMSO-70\%}$, PNASC$_{DMSO-80\%}$, PNASC$_{DMSO-90\%}$, and PNASC$_{DMSO-100\%}$). The PL spectrum of

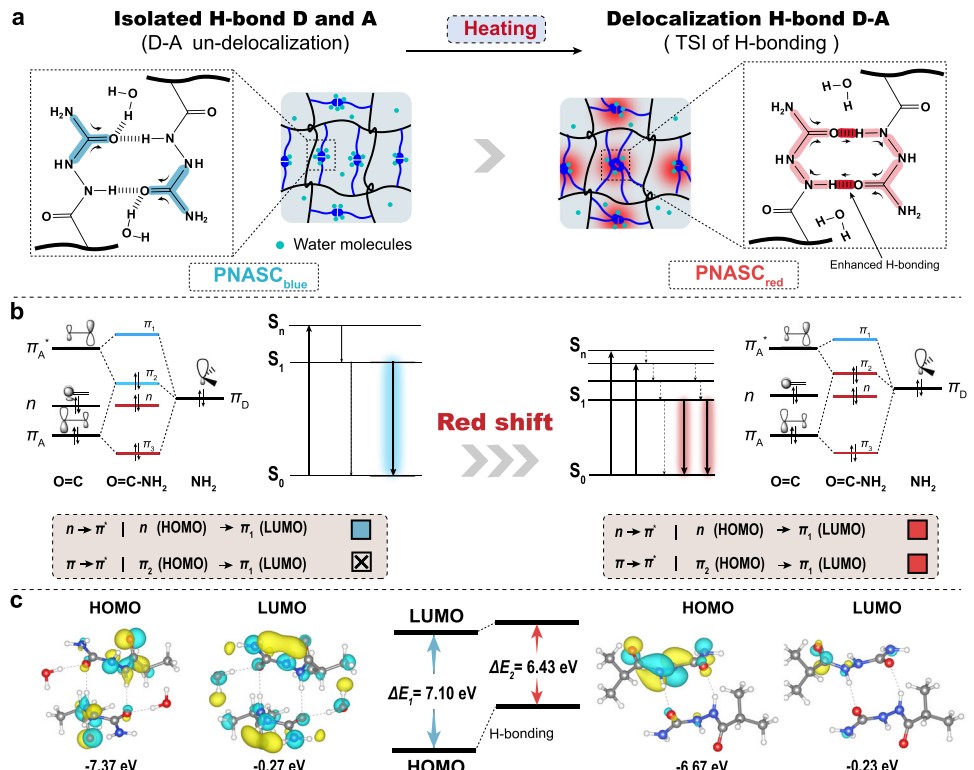

**Fig. 4 | Luminescent mechanism of hydrogels. a** Schematic diagram of the process of conformational change of PNASC hydrogels. PNASC$_{25°C}$ hydrogel is named PNASC$_{blue}$, and PNASC$_{100°C}$ hydrogel is named PNASC$_{red}$. The blue and red shadings on the molecular structures represent the emission fluorescent colors and electron delocalization extent of corresponding polymer configurations, respectively. **b** Schematic illustration of red shift of emission spectra. Red and blue squares represent the fluorescent colors emitted by photons, respectively. **c** HOMO and LUMO orbits.

PNASC$_{DMSO-70\%}$ hydrogel, which possessed the highest interchain H-bond density, exhibited the most distinct absorption peaks and greatest transmittance at approximately 365 nm and 550 nm (Supplementary Fig. 19, Fig. 3a(iii), 3a(iv)). Moreover, as-PNASC$_{DMSO-70\%}$ was further treated at 100 °C for various durations, and the resulting materials were named PNASC$_{12h}$, PNASC$_{24h}$, PNASC$_{48h}$, and PNASC$_{72h}$. The fluorescence of the hydrogels shifted from the blue to the red region and remained stable after approximately 48 hours of processing (Fig. 3b, Supplementary Fig. 20, Supplementary Fig. 21). In addition, we investigated the correlation between the polymer fractions and fluorescence intensity (Fig. 3c, Supplementary Fig. 22, Supplementary Fig. 23). The results suggested that the fluorescence intensity gradually increased with increasing polymer fraction from PNASC$_{10\%}$ to PNASC$_{25\%}$, which was attributed to the increase in cluster density. Here, x% represents the percentage of monomer relative to the total mass, heating time is 48 hours.

**Luminescence mechanism of the hydrogels**

As illustrated in Fig. 4a, the interactions between PNASC chains primarily involved the formation of H-bonds between the −C=O moiety in the urea group and the −N−H moiety in the amide. For the PNASC$_{25°C}$ hydrogel, which emitted blue fluorescence (PNASC$_{blue}$), the hydrogen bonds that formed between −C=O and −N−H experienced interference from hydrated water molecules in the hydrogel matrix, which resulted in an isolated H-bond D−A state. In contrast, the PNASC$_{100°C}$ hydrogel that emitted red fluorescence (PNASC$_{red}$) possessed strong H-bonding interactions to limit the hydration of water molecules by thermodynamic promotion, which led to conformational rigidity and the formation of tight clusters. As a result, the clustered structures promoted the formation of large electron delocalization via TSI involving electron-rich groups. Thus, strong H-bonds decreased the distance between the −C=O and −N−H moieties to induce intermolecular

electron delocalization, and through-space charge transfer could occur within every H-bond D-A unit under the influence of water-independent H-bonding (Supplementary Fig. 24). Besides, thermodynamic insight further confirmed that the fluorescent hydrogels tended to form clustered structures to minimize the system energy by conformational variation ($\Delta G = G_2 - G_1 < 0$) (Supplementary Table 1). In addition, it is also worth noting that a substantial number of heteroatoms participate in hydrogen bond formation in the hydrogel, which also plays a critical role in building a favorable environment for cluster luminescence. Besides, a small fraction of heteroatoms containing lone-pair electrons that do not form strong H-bonds may also participate in n−n, n−π interactions of TSI. So, the contribution of this part should not be overlooked (Supplementary Fig. 25)

Furthermore, the analysis of the frontier molecular orbitals also supported the abovementioned results (Fig. 4b). The electron-donating groups -NH and -NH$_2$ possessed more lone pairs of π electron orbitals, denoted as $\pi_d$. The $\pi_A$ and $\pi_A$* orbitals of −C=O interacted with $\pi_D$ to generate three orbitals, $\pi_1$, $\pi_2$, and $\pi_3$, where $\pi_1$ is a strong bonding orbital and $\pi_3$ is a strong antibonding orbital. For PNASC$_{blue}$, the energy level gap between $\pi_D$ and $\pi_A$ of −C=O was smaller than that between $\pi_D$ and $\pi_A$*. According to the principle of energy similarity, the interactions between the $\pi_D$ and $\pi_A$ orbitals of −C=O were stronger than those between $\pi_D$ and $\pi_A$*. Thus, $\pi_2$ is a weak antibonding orbital. In addition, given the multiple electron-donating group interactions (-NH and -NH$_2$), it was easier for electrons to undergo $n \to \pi^*$ ($n \to \pi_1$) transitions than $\pi \to \pi^*$ transitions. However, for PNASC$_{red}$, the enhancement of electron delocalization improved the electron-pushing ability of -NH and -NH$_2$ to generate higher lone pair electron orbitals in $\pi_D$. The energy level gap between the $\pi_D$ and $\pi_A$ orbitals of −C=O was larger than that between $\pi_D$ and $\pi_A$*. In this case, $\pi_2$ was considered a weak bonding orbital. The electrons could undergo not only $n \to \pi^*$ ($n \to \pi_1$) transitions but also $\pi \to \pi^*$ ($\pi_2 \to \pi_1$) transitions.

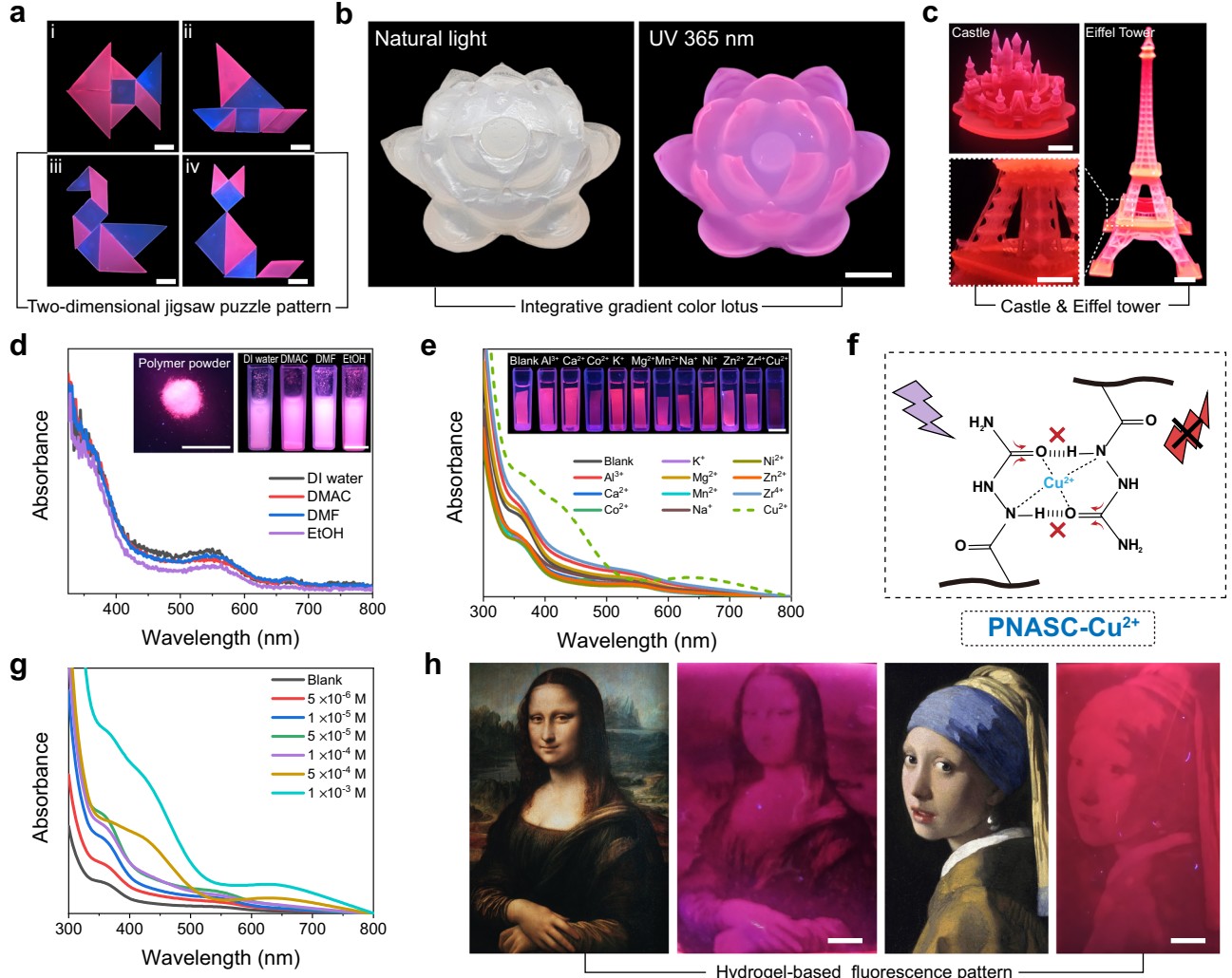

**Fig. 5 | Fluorescent hydrogel architectures and responsiveness. a** Photos of a tangram puzzle of fluorescent hydrogels under the UV 365 nm. Blue is PNASC_blue hydrogel, red is PNASC_red hydrogel. The scale bar is 1 cm. **b** Gradient fluorescent lotus-shaped hydrogel under natural light and UV 365 nm, respectively. The scale bar is 1 cm. **c** Hydrogel castle and Eiffel tower. The scale bar is 5 mm. **d** Absorbance spectra of PNASC_red powder in DI water, DMAC, DMF, and EtOH, respectively, 0.1 g/ mL. The scale bar is 1 cm. **e** Fluorescent responses of PNASC_red hydrogel to different metal ions. The scale bar is 1 cm. **f** Detection mechanism of Cu$^{2+}$. The hydrogel after coordination of PNASC_red with Cu$^{2+}$ is named PNASC-Cu$^{2+}$. **g** Absorbance spectra of PNASC_red hydrogels at different Cu$^{2+}$ concentrations. **h** Hydrogel-based fluorescence painting with high contrast and distinguished color gradient. The scale bar is 5 mm.

In particular, the electron-donating group and greater electron delocalization increased the $\pi_2$ (HOMO) energy level to be higher than the $\pi_1$ (LUMO) energy level. The results of theoretical calculations revealed that PNASC_red with intermolecular TSI possessed a smaller HOMO/LUMO gap ($\Delta E_2 = 6.43$ eV) than PNASC_blue ($\Delta E_1 = 7.10$ eV), which was essential for the red fluorescence of the PNASC hydrogel (Fig. 4c, Supplementary Table 2, Supplementary Data 1).

### Red fluorescent hydrogel structures and responsiveness

The proposed red fluorescent hydrogel system was adapted for high-precision structured manufacturing through digital light processing (DLP) 3D printing technology[37,38]. As depicted in Fig. 5a-c, we readily constructed diverse fluorescent hydrogel architectures, such as two-dimensional jigsaw puzzle patterns, an integrative gradient color lotus (only heating petals), and even a complex Eiffel Tower and sophisticated castle architectures. Moreover, the hydrogel and corresponding polymer powders exhibited good stability and compatibility in various solvents, including DI water, N,N-dimethylacetamide (DMAC), N,N-dimethylformamide (DMF), and ethanol (EtOH), without losing their red fluorescence performance (Fig. 5d, Supplementary Fig. 26). Moreover,

the reversible addition-fragmentation chain transfer polymerization (RAFT)-mediated modification of the hydrogel particles with polyvinyl pyrrolidone resulted in desirable hydrophilicity, allowing stable dispersion in water for several hours (Supplementary Fig. 27). In addition, Cell Counting Kit-8 (CCK-8) cell viability tests were also conducted to test the biocompatibility of the PNASC_100°C hydrogel. As shown in Supplementary Fig. 28, the proposed fluorescent hydrogels maintained a cell viability of 95.30 ± 2.93% after 7 days, demonstrating favorable cell biocompatibility.

The groups (O=C−NH−) with abundant lone-pair electrons in the PNASC network possessed strong coordination ability with metal ions that possess $p$ orbitals. As shown in Fig. 5e, f, the fluorescent emission of PNASC_100°C was selectively quenched by Cu$^{2+}$ due to the strong coordination between Cu$^{2+}$ and O=C−N. Therefore, the visual recognition and detection of metal ion (Cu$^{2+}$) concentrations could be achieved based on the visible color change together with the variation in the emission intensity of the fluorescent hydrogels (Fig. 5g). Relying on the controllable and selective fluorescent quenching effect of Cu$^{2+}$, we subtly fabricated hydrogel patterns with different crosslinking densities by grayscale DLP 3D printing[39,40] to regulate Cu$^{2+}$ coordination.

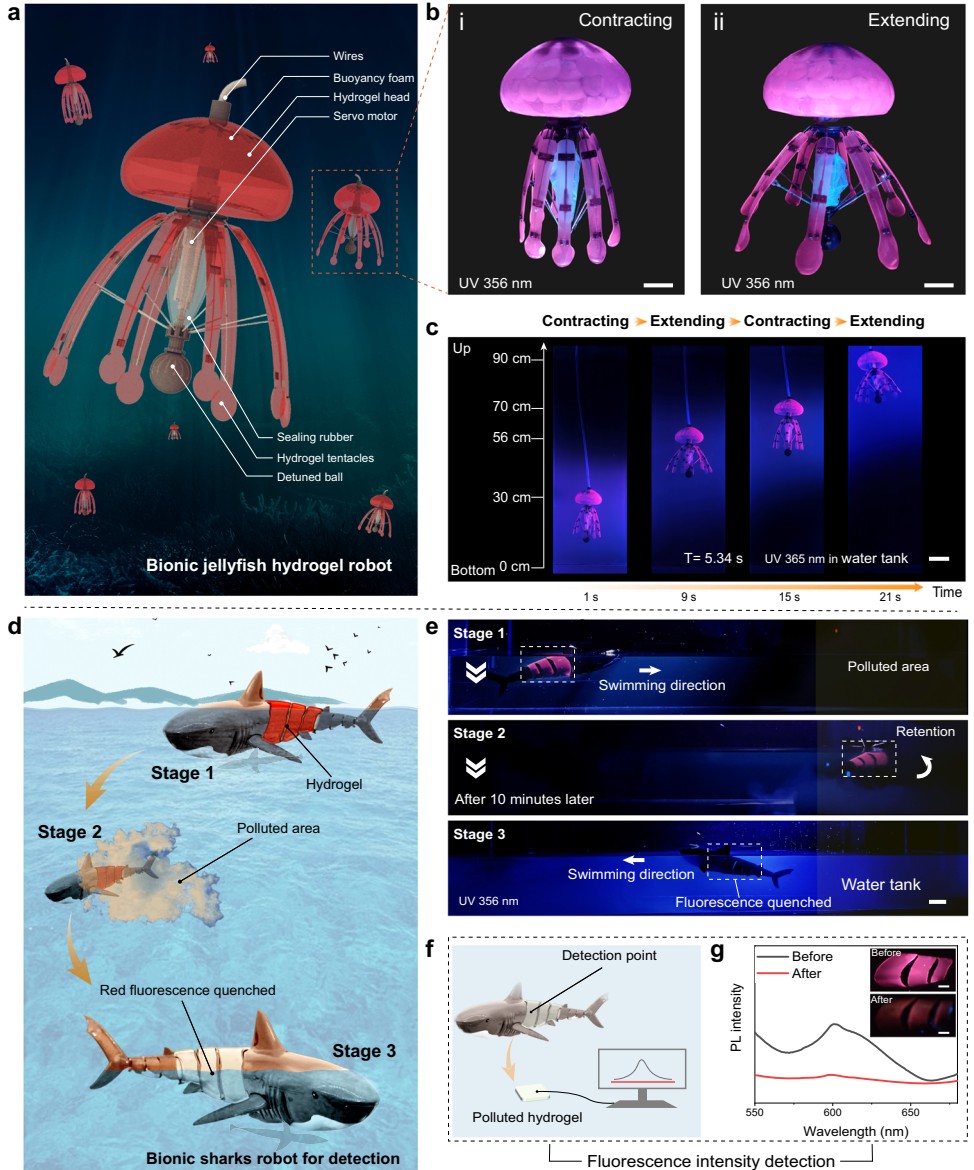

**Fig. 6 | Underwater fluorescent hydrogel actuators and applications.**
**a** PNASC$_{100°C}$ hydrogel as fluorescent components for a bionic jellyfish underwater robot. **b** Digital photographs of the bionic jellyfish robot extending and contracting under Ex 365 nm. The scale bar is 3 cm. **c** Snapshot images of the swimming trajectory of the bionic jellyfish robot. T represents the time of one cycle of contracting and extending. The scale bar is 10 cm. **d** Schematic diagram of the contaminant detecting process by a fluorescent electric hydrogel-shark. **e** Snapshot images of contaminant detecting process of the fluorescent electric hydrogel-shark. The scale bar is 5 cm. **f** Schematic diagram of the hydrogel detection process and **g** its corresponding emission spectra excited at 365 nm. The inset is a digital photograph of a hydrogel before and after being polluted under 365 nm light. The scale bar is 2 cm.

The low-crosslink-density hydrogel domains clearly displayed more rapid diffusion of $Cu^{2+}$ than the high-crosslink-density domains (Supplementary Fig. 29). As a result, the heterogeneity of $Cu^{2+}$ coordination in the hydrogel networks could drive the difference in fluorescence intensity, contributing to fluorescent hydrogel paintings with good resolution and color contrast (Fig. 5h).

### Underwater fluorescent actuators and applications

These fluorescent hydrogels hold great promise for application as soft materials for illuminated components and devices in a variety of scenarios, including contaminant detection and soft robot construction. To demonstrate such potential, we designed and fabricated a bionic fluorescent jellyfish swimming robot by 3D printing a robot head and tentacles with our PNASC$_{100°C}$ hydrogel (Fig. 6a). Under the driving force of a servo motor in the central position, the periodic extension and contraction of support rods drove the extension and contraction of the hydrogel tentacles in the surrounding water (Fig. 6b, Supplementary Fig. 30). As shown in Fig. 6c, Supplementary Movies 2 and 3, the cyclic motion of the tentacles generated a propulsive thrust in water, allowing the hydrogel jellyfish to swim underwater. The fluorescent hydrogel exhibited good mechanical robustness in aquatic environments, which was attributed to the dense polymer network generated by H-bonds.

On the other hand, engineering fluorescent hydrogel materials for use in underwater robotics for the rapid recognition of heavy metal ions enabled applications such as sensing pollution in water. As shown in Fig. 6d, we also designed another underwater electrically driven bionic robotic shark as a fluorescent sensing device equipped with our fluorescent hydrogel skin (Supplementary Fig. 31). To simulate the detection of harmful substances in aquatic

environments by the fluorescent sharks, we used $Cu^{2+}$ as a contaminant to validate the detection potential. The red fluorescence of the hydrogel skin of the robot was significantly attenuated after swimming in polluted water for several minutes (Fig. 6e, Supplementary Movie 4). Finally, the PL spectra of the contaminated fluorescent hydrogels was used to quantitatively analyze the concentrations of pollutants in contaminated water based on the relationships mentioned above (Fig. 6f and g).

## Discussion

In summary, we developed a thermodynamics-driven strategy to achieve delocalized H-bonding-induced nonconventional red fluorescence emission in hydrogels. In this case, the fluorescence of supramolecular multiple H-bonding hydrogels with robust networks was extended to the red region through a simple heating process. The red shift of the fluorescence was attributed to the heat-induced rearrangement and aggregation of polymer chains, which promoted the formation of large through-space interaction structures by strong H-bonding interactions and restrained the hydration of water molecules. The electron-rich groups in the hydrogel extended the electron delocalization through intermolecular charge transfer to achieve stable red fluorescence emission in a water environment by various heteroatom-involved interactions (including heteroatom-involved hydrogen bonds and $n$-$n/\pi$ TSI). The fluorescence intensity and region were flexibly regulated by protocol parameters, including the heating temperature, duration, and polymer fractions on demand. This method also exhibited several advantages, including a green synthetic route, high efficiency, easy implementation, and customizable processability. In particular, the developed nonconventional long-wavelength red fluorescent hydrogels exhibited desirable biocompatibility, fluorescent stability, mechanical robustness and detection of metal ions, making them very attractive for application in the fields of biology, medicine, detection, and soft robotics.

## Methods

### Synthesis of N-acryloylsemicarbazide

Semicarbazide hydrochloride (6.25 g) 6 mL of DI water, 33.6 mL of a 2 M $K_2CO_3$ solution (cooled), and 18 mL of cold diethyl ether were added sequentially to a 100 mL round-bottom flask. To the above solution, with stirring at 0 °C for approximately 4 h, 5.7 g of acryloyl chloride in 24 mL of diethyl ether was added dropwise. Afterward, the generated white precipitate was collected by filtration to obtain the crude product. The crude product was dissolved in DI water and stirred for 1 h at 95 °C; subsequently, the undissolved precipitate was removed by centrifugation, and the supernatant was freeze-dried to obtain the final NASC monomer[36]. The average yield of NASC was ~66%.

### Preparation of fluorescent hydrogels

To prepare 10, 15, 20, and 25 wt% NASC solutions, the NASC monomer and LAP (0.3 wt% of the NASC monomer) were dissolved in a mixed solvent containing dimethyl sulfoxide (DMSO) and deionized (DI) water. Then, the NASC solution was stirred in a nitrogen atmosphere for 60 min. The as-hydrogels were first cured by UV light (405 nm). Then, the as-prepared hydrogel samples were thoroughly immersed in DI water at 25 °C, 45 °C, 65 °C, 85 °C, and 100 °C for different durations.

### Preparation of the fluorescent particle solution

The NASC monomer (25 wt%), 4-cyano-4-(((ethylthio)carbonothioyl) thio) pentanoic acid (RAFT agent, 0.1 wt% of the NASC monomer), and LAP (0.3 wt% of the NASC monomer) were dissolved in a mixed solvent of DMSO and DI water (the ratio of DMSO to DI water was 7:3 (wt/wt)). The NASC solution was stirred in a nitrogen atmosphere for 60 minutes and cured using UV-405 light for 20 s to produce the resulting hydrogel. Next, the red fluorescent hydrogel was obtained by immersing the sample in deionized water at 100 °C for 48 h, after which it was freeze-dried and ground into a powder. The red fluorescent polymer powder was immersed in a solution of 0.5 M N-vinyl-2-pyrrolidone (NVP) in dimethylformamide (DMF). Then, 1 wt% ammonium persulfate was added, and the mixture was stirred magnetically under a nitrogen atmosphere at 60 °C for 6 h. Finally, hydrophilic fluorescent particles were obtained by centrifugation, followed by dispersion in deionized water, resulting in a red fluorescent particle solution.

### Characterization instruments and methods

Fourier transform infrared (FT-IR) spectra were recorded with a TFS-66 V/S spectrometer (Bruker) using the KBr-disk method. Nuclear magnetic resonance (NMR) spectra were recorded by means of a Bruker Avance Neo 400WB. The absorbance and transmittance spectra were recorded with a Shimadzu UV-2600. The PL spectra and photoluminescence quantum yield (PLQY) were recorded by means of a Fluorolog-3 fluorescence spectrometer (HORIBA Instruments, UK) and an Absolute PL C9920-02G instrument. The time-resolved fluorescence decay curves were recorded by means of an FLS1000 instrument (Edinburgh). The micromorphologies of the hydrogels were examined via scanning electron microscopy (SEM; Phenom ProX, Netherlands) at a voltage of 10-30 keV. Specifically, cross-sections of the hydrogel samples were obtained by liquid nitrogen quenching followed by mechanical bending.

### Water content measurement

The water content of the hydrogels was determined from the weight difference of the hydrogels before and after water removal. The water content was expressed as a percentage of the hydrogel's total weight and was calculated using the following equation:

$$\text{water content} = \frac{W - W_0}{W} \times 100\%$$

where $W$ and $W_0$ are the weights of the hydrogel before and after drying, respectively.

### Tensile testing

A uniaxial tensile test was performed on an electronic universal material testing machine equipped with a 500 N load cell (EZ-Test, Shimadzu, Japan). The hydrogels, which had rectangular shapes measuring 35 mm in length, 5 mm in width, and 0.8-1 mm in thickness, were utilized for the tensile measurements. The tests were conducted in water at room temperature (RT) with a crosshead velocity of 100 mm/min. Each point was measured for at least three samples.

### Electrochemical measurements

Electrochemical measurements were carried out with a CHI 660E electrochemical workstation with a three-electrode system at room temperature. The hydrogel (5.0 mm × 2.0 mm × 0.5 mm) was used as the working electrode (WE), a Pt wire was used as the counter electrode (CE), and a Ag/AgCl/sat was used. The KCl electrode was used as the reference electrode (RE). A light source system (PL-FX300HU) was used as the irradiation source, and the distance between the light source and the electrode surface was fixed at 10 cm. Photoelectrochemical detection was conducted at a constant potential of 0.7 V vs. Ag/AgCl in 0.01 M $K_4Fe(CN)_6$ and 0.1 M KCl.

### Fluorescent hydrogels and polymers in different solutions

The hydrogel was freeze-dried and ground into powder. 0.5 g $PNASC_{100°C}$ powder was added to 5 mL of DI water, DMF, DMAC, or EtOH. $PNASC_{100°C}$ hydrogels (30 mm long × 5 mm wide × 0.8-1 mm thick) were added to DI water, DMF, DMAC, or EtOH for 4 h, respectively. Then, the absorbance spectra were recorded by a Shimadzu UV-2600 spectrophotometer.

## Metal ion detection

Aqueous solutions (1 mM/L) of metal ions ($Al^{3+}$, $Ca^{2+}$, $Co^{2+}$, $K^+$, $Mg^{2+}$, $Mn^{2+}$, $Na^+$, $Ni^+$, $Zn^{2+}$, $Zr^{4+}$, and $Cu^{2+}$) and $PNASC_{100°C}$ hydrogels (30 mm long × 5 mm wide × 0.8 ~ 1 mm thick) were prepared in advance. The $PNASC_{100°C}$ hydrogels were added to 20 ml of each metal ion solution for 12 h. The absorbance spectra of the hydrogels were recorded to determine the selectivity for detecting metal ions.

In addition, a series of aqueous $Cu^{2+}$ solutions with different concentrations (5-1000 μM) were prepared. $PNASC_{100°C}$ hydrogels (30 mm long × 5 mm wide × 0.8-1 mm thick) were added dropwise to 20 mL of each $Cu^{2+}$ solution, after which the absorbance intensity variation of the hydrogels was recorded.

## Fluorescent pattern printing

The hydrogel patterns were fabricated by a commercial DLP 3D printer with a 405 nm projector. Specifically, the photographs to be printed were converted into grayscale images. The patterns of the as-hydrogel were obtained from grayscale hydrogels through ultraviolet exposure. The curing time of each layer (5 s) and the layer thickness (0.05 mm) remained constant for all the experiments. The as-prepared hydrogels were thoroughly immersed in DI water at 100 °C for 48 h. Then, the hydrogel patterns were obtained by immersing the samples in a $Cu^{2+}$ solution (0.1 mM) for 20 s.

## Cell viability assay

L929 cells (Item No.: CL-0137, Procell Life Science&Technology Co., Ltd) were used for cell experiments (Supplementary Table 3). $PNASC_{100°C}$ hydrogel samples (3.125, 6.25, and 12.5 mg) were soaked in 1 mL of culture solution for 24 h. After culturing for 1 day, 4 days, or 7 days, the medium was removed, and the wells were washed three times with PBS. Cell viability was assessed using a cell counting kit-8 IV08 (CCK-8) following the manufacturer's instructions. Specifically, 100 μl of CCK-8 solution was added to each well of a 96-well plate, and the cells were incubated at 37 °C for 2 h. The absorbance intensity of each sample was measured at a wavelength of 450 nm using a microplate reader (SPARK 10 M, TECAN, Switzerland). The experiments were repeated three times to ensure accuracy.

The blank group included wells without medium or cells, and the control group included wells with medium containing only cells without samples. The optical density (OD) values of the blank group, control group, and experimental group were denoted as $OD_{bla}$, $OD_{con}$, and $OD_{exp}$, respectively. The cell survival rate was calculated as follows:

$$\text{cell viability} = \frac{OD_{exp} - OD_{bla}}{OD_{con} - OD_{bla}} \times 100\% \qquad (2)$$

## Statistics and reproducibility

EXCEL was used to implement the statistical analysis. Preprocessing of data was transformed. For Figs. 2b, c, d, e, f, h, i, 3a(iii), (iv), b, c, 5d, e, g, and 6g, Supplementary Figs. 13, 15, 19–23, experiments have been independently tested at least 3 times with similar data. For Fig. 2g, Supplementary Figs. 6, 14, 17, and 28, the values were expressed as mean ± standard deviation, each experiment was repeated independently with three times.

## Reporting summary

Further information on research design is available in the Nature Portfolio Reporting Summary linked to this article.

## Data availability

The authors declare that data supporting the findings of this study are available within the paper and its supplementary information files. The data is available from the authors on request. Source data are provided with this paper.

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

## Acknowledgements

X.W. thanks the following programs for the financial support: the National Key R&D Program of China (2022YFB4600101), the National Natural Science Foundation of China (52205228, 52175201and 52005484), the Strategic Priority Research Program of the Chinese Academy of Sciences (XDB 0470303), the Major Program (ZYFZFX-2) and the Cooperation Foundation for Young Scholars (HZJJ23-02) of the Lanzhou Institute of Chemical Physics, CAS, and the Western Light Project of CAS (xbzg-zdsys-202007), the Taishan Scholars, and the Oasis Scholar of Shihezi University.

## Author contributions

X.W. and P.J. co-designed and convinced the project. J.W. designed and performed all experiments. X.W. supervised the project. Y.W. helped with all electrochemical measurements. J.W., P.J. and X.W. wrote the manuscript. X.J. and F.Z. provided assistance in data analysis and revision. All authors contributed to the analysis and discussion of the data.

## Competing interests

The authors declare no competing interests.
