## [Peer Review File · Nature Communications]

Multiple hydrogen-bonding induced nonconventional red fluorescence emission in hydrogelsREVIEWER COMMENTS

Reviewer #1 (Remarks to the Author):

This study (NCOMMS-23-56999-T) introduces an innovative thermodynamics-driven method to achieve unique red fluorescence in hydrogels, relying on delocalized H-bonding. The hydrogels, with robust networks, demonstrate a shift to red fluorescence through a straightforward heating process. This red-shifted emission arises from the heat-induced rearrangement and aggregation of polymer chains, fostering the creation of extensive through-space conjugation structures via strong H-bonding. Electron-rich groups in the hydrogel extend electron delocalization, enabling stable red fluorescence emission in water. Moreover, their method offers flexibility, allowing the regulation of emission intensity and region through changing parameters (i.e., temperature, heating duration, and polymer fractions). The resulting nonconventional red fluorescent hydrogel exhibits favorable properties for diverse applications in biomedicine, detection, and soft robotics, addressing crucial gaps in clustering-triggered emission (CTE) research and indicating promising directions for through-space interaction (TSI) studies. Thus, I am enthusiastic about recommending the acceptance and publication of this manuscript in Nature Communications after minor modifications:

- (1) For this non-aromatic hydrogel system, I suggest the term, “through-space interaction” (TSI), is more appropriate than “TSC”. The words “spatial conjugation” and “through-space conjugates” should also be replaced. Additionally, more recent studies on TSI may also be considered for citation.
- (2) Their explanation of the luminescence mechanism in Fig. 1 and Fig. 4 is not comprehensive enough. In my opinion, thermodynamic promotion may also lead to strong intra-chain or inter-chain n-n TSI derived from a large of heteroatoms with lone pairs because of the enhancement of conformational rigidity and the formation of tight clusters. Please discuss this in the main text fully with experimental data or theoretical calculations.
- (3) As stated by the authors, “with the heating temperature increased from 25 °C to 100 °C, the PLQY of the fluorescent hydrogels increases from 1.4 ± 0.20 % to 3.26 ± 0.06 % excited at 365 nm, and from 0 % to 2.73 ± 0.15 % excited at 530 nm”. However, I think that the current QY measurement technology is not advanced enough for the deviation range to be so small. Please keep the values of QY in single digits.
- (4) The fluorescent phenomena provided by the authors are very bright. Please provide the wattage of the light source.
- (5) Please provide the time-resolved fluorescence decay curves of these materials to determine the nature of the luminescence.
- (6) In Fig. 4c, there are no obvious D-A structures from the distribution of HOMO-LUMO orbits. Please provide more evidence to prove the existence of delocalization D-A.

Reviewer #2 (Remarks to the Author):

In this paper, Wang et al. propose a thermodynamics-driven strategy to achieve the delocalized H-bonding induced nonconventional red fluorescent emission in hydrogel. They discussed the fundamental mechanism of this emergent phenomenon in fluorescent hydrogel. Importantly, these contributions show significant implications for deeply understanding the inherent clustering-triggered emission (CTE) when forming the spatial conjugation in water surroundings. Moreover, the authors also verified the feasibility of 3D-printed red fluorescent hydrogels in applications such as fluorescence detection and soft robotics by grayscale digital light processing (g-DLP) technology. This work bridges an obvious knowledge gap and provide a new insight in understanding the inherent clustering-triggered emission (CTE) when forms the spatial conjugation in water surrounding. Therefore, it can be considered for publication in Nature Communications, but some minor comments are still required before publication.

1. In Fig. 2d, the PNASC100 °C hydrogel also exhibits a blue light emission region at approximately 425 nm, please provide additional explanation.
2. The fluorescent color depends on the temperature of post-heating. It can be found that the PNASC65°C hydrogel retained blue fluorescent properties, while the PNASC85°C hydrogel exhibited a sudden transition to red fluorescence. It's a large wavelength shift from blue to red. The authors should further discuss this process. And meanwhile, we are curious as to why hydrogels only show the isolated wavelength shift (i.e., blue or red) instead of a transition state for other middle wavelength.
3. For the solvent stability of hydrogels, the paper only verifies the fluorescence stability of polymer powders. However, the fluorescence stability of hydrogel was not verified in detail. The supplementation is recommended.
4. In Supplementary Fig. 5, the elongation at break of the PNASC100°C hydrogel decreases after heating. What's the reason for this?
5. For the Cell viability assay, the article does not mention the type of cells used, and additional clarification is needed to declare.

Reviewer #3 (Remarks to the Author):

Reviewer #4 (Remarks to the Author):

Overall the article is poorly written and requires an extensive spelling and grammar check.

p. 3: No reference is made of a crosslinker. The authors refer to the use of N-acryloylsemicarbazide in the presence of an initiator to create a hydrogel. However, starting from this monomer, no hydrogel networks can be formed. Instead only linear polymer chains can be formed with lack of stability in the presence of a solvent. Furthermore, the authors should elaborate on possible side reactions occurring in DMSO as solvent when performing a radical-mediated polymerization.

Furthermore, the authors should evidence hydrogel stability by referring to gel fractions obtained following polymerization. I would also recommend to analyze the amount of unreacted double bonds with HR-MAS NMR spectroscopy to complement gel fraction assays.

Materials & Methods

p.17: What yield was obtained for the monomer synthesis?

p.20: How was DLP optimized? What was the resolution/CAD-CAM mimicry observed? It is surprising that the authors did not require photo-absorber for DLP (since this is typically applied in DLP to ensure appropriate resolution).

The materials & methods section reports on cell viability assays, but no results are mentioned in the article body regarding the execution of such experiments.

Response to Reviewers

Reviewer #1 (Remarks to the Author):

This study (NCOMMS-23-56999-T) introduces an innovative thermodynamics-driven method to achieve unique red fluorescence in hydrogels, relying on delocalized H-bonding. The hydrogels, with robust networks, demonstrate a shift to red fluorescence through a straightforward heating process. This red-shifted emission arises from the heat-induced rearrangement and aggregation of polymer chains, fostering the creation of extensive through-space conjugation structures via strong H-bonding. Electron-rich groups in the hydrogel extend electron delocalization, enabling stable red fluorescence emission in water. Moreover, their method offers flexibility, allowing the regulation of emission intensity and region through changing parameters (i.e., temperature, heating duration, and polymer fractions). The resulting nonconventional red fluorescent hydrogel exhibits favorable properties for diverse applications in biomedicine, detection, and soft robotics, addressing crucial gaps in clustering-triggered emission (CTE) research and indicating promising directions for through-space interaction (TSI) studies. Thus, I am enthusiastic about recommending the acceptance and publication of this manuscript in Nature Communications after minor modifications:

Reply: We sincerely thank you for your time and efforts in reviewing our paper. We have revised the manuscript carefully based on your good comments and valuable suggestions. All modifications are shown in red in the revised manuscript.

(1) For this non-aromatic hydrogel system, I suggest the term, “through-space interaction” (TSI), is more appropriate than “TSC”. The words “spatial conjugation” and “through-space conjugates” should also be replaced. Additionally, more recent studies on TSI may also be considered for citation.

Reply: Thank you for your good comments. We investigated the relevant literatures about “through-space interaction” (TSI) and “through-space conjugates”. Given the character of the non-aromatic supramolecular hydrogel system in this work, we agree with that TSI is the natural mechanism of this red fluorescent hydrogel instead of TSC. We have replaced TSC with TSI one by one throughout the manuscript. Besides, we also added the discussion about TSI by citing the recent relevant literature.

In the revised main text, we have added the relevant literature about TSI as follows:

- “25. Liu, J. *et al.* Through-Space Interaction of Tetraphenylethylene: What, Where, and How. *J. Am. Chem. Soc.* **144**, 7901-7910 (2022).
26. Zhang, J. *et al.* Secondary through-space interactions facilitated single-molecule white-light emission from clusteroluminogens. *Nat. Commun.* **13**, 3492 (2022).
27. Chu, B. *et al.* Enabling nonconjugated polyesters emit full-spectrum fluorescence from blue to near-infrared. *Nat. Commun.* **15**, 366 (2024).”

(2) Their explanation of the luminescence mechanism in Fig. 1 and Fig. 4 is not comprehensive enough. In my opinion, thermodynamic promotion may also lead to strong intra-chain or inter-chain n-n TSI derived from a large of heteroatoms with lone pairs because of the enhancement of conformational rigidity and the formation of tight clusters. Please discuss this in the main text fully with experimental data or theoretical calculations.

Reply: Thanks for your professional comments. According to the TSI mechanism, one of the key factors leading to cluster luminescence is spatial interaction, which includes *n-n*, *n- π* , *π - π* , hydrogen bond (H-bond) interactions, and so on. Among of them, the H-bonding interaction plays an important role in cluster luminescence. Specifically, H-bonding can lead to fluorescent properties through electron delocalization, and meanwhile strong H-bonding originated from heteroatoms with lone pairs also has the potential for fluorescence emission through spatial interactions. Therefore, we further investigate that the clustering-triggered emission effect is dominated by electron delocalization caused interchain strong multiple H-bonding interactions and not the aggregation of heteroatoms that's causing it. As shown in **Supplementary Fig. 19**, we illustrated this by employing as-PNASC_{DMSO-0%} hydrogels with severe phased-separation structures aggregated by intrachain multiple H-bonding (**Supplementary Fig. 10**). Its tight clusters may contain interactions such as *n-n* and *n- π* . The as-PNASC_{DMSO-0%} was then heated as the proposed protocol and named as PNASC_{DMSO-0%}. It can be found that there is no obvious red fluorescent emission yet only a weak bit of red presumably due to a small amount of interchain hydrogen bonding after heating.

Supplementary Fig. 19 | (a) Digital photos of as-PNASC_{DMSO-0%} and PNASC_{DMSO-0%} hydrogels under natural light and UV light (365 nm). (b) Interaction of clusters in PNASC_{DMSO-0%} hydrogels. The as-PNASC_{DMSO-0%} was heated at 100°C for 48 h and named PNASC_{DMSO-0%}. The scale bar is 5 mm.

In addition, we previously discussed that solvent modulation of polymer conformation leads to hydrogels with different interchain hydrogen bonding content in the main text. As shown in Fig. 3a, there is a strong correlation between fluorescence performance and polymer conformation of hydrogel after heating. The PNASC_{DMSO-70%} hydrogel with the highest interchain H-bond density exhibits the most distinct absorption peaks and transmittance at approximately 365 nm and 550 nm. It is also suggested by the PL spectrogram (Fig. 3aiii and iv). The above-mentioned results reflect the dominance of interchain H-bonding for generating fluorescent emission.

Fig. 3a | Effect of different solvent ratios on fluorescent hydrogels. (i), Schematic diagram of the relationship between different solvent ratios and the relative content of inter-/intra-chain H-bonds in polymers. (ii), Tensile-strain curves of as-PNASC_{DMSO-0%}, as-PNASC_{DMSO-30%}, as-PNASC_{DMSO-50%}, as-PNASC_{DMSO-70%}, and as-PNASC_{DMSO-100%}. Emission spectra images of PNASC_{DMSO-50%}, PNASC_{DMSO-70%}, PNASC_{DMSO-80%}, and PNASC_{DMSO-100%}.

PNASC_{DMSO-90%}, and PNASC_{DMSO-100%} of excited at (iii) 365 nm and (iv) 530 nm. Finally, as question 3 raised by reviewer 2, although the PNASC_{100°C} hydrogel remained good stability and still generate red fluorescent emission in various solvents, such as DI water, DMAC, DMF, and EtOH, the red fluorescence of PNASC_{100°C} hydrogel gradually disappeared in DMSO solvent. It's well known that DMSO has strong ability to break the H-bonds (**Fig. R1**). The result also verifies that the red fluorescence property of hydrogel is dominated by interchain hydrogen bonding actions.

Fig. R1 | Photos of PNASC_{100°C} hydrogel in DMSO. The scale bar is 1 cm.

According to the above experiments and analyses, we surmised that fluorescent properties of hydrogels mainly derive from the H-bond D-A structure between polymer chains by strong multiple H-bonding interactions under heat-induced rearrangement and aggregation, rather than n-n, n- π interaction and so on.

We added this result in **Supplementary Fig. 19**, and more explanation has been added into “**Luminescent mechanism of hydrogels**” section in the revised manuscript. The revision made was as follows:

“In addition, to further illustrate that strong multiple H-bonding interactions, rather than other interactions (such as n-n, n- π interactions), had the dominant effect on the fluorescent properties, as shown in **Supplementary Fig. 19**, we employed the as-PNASC_{DMSO-0%} hydrogel with severe phase separation structures originating from intrachain multiple H-bonding. Its tight clusters contained many n or π interactions in the -C=O and -N-H moieties. However, there was no obvious red fluorescence but only a small amount of red fluorescence, presumably due to the small amount of interchain hydrogen bonding after heating.”

(3) As stated by the authors, “with the heating temperature increased from 25 °C to 100 °C, the PLQY of the fluorescent hydrogels increases from $1.4 \pm 0.20\%$ to $3.26 \pm 0.06\%$ excited at 365 nm, and from 0 % to $2.73 \pm 0.15\%$ excited at 530 nm”. However, I think that the current QY measurement technology is not advanced enough for the deviation range to be so small. Please keep the values of QY in single digits.

Reply: Thanks for your helpful advice. We have revised the values of PLQY to single digits. The modified sentence is “the photoluminescence quantum yield (PLQY) of the fluorescent hydrogels increased from $1.4 \pm 0.2\%$ to $3.3 \pm 0.1\%$ upon excitation at 365 nm and from 0% to $2.7 \pm 0.2\%$ upon excitation at 530 nm”, which has been corrected in the revised manuscript.

(4) The fluorescent phenomena provided by the authors are very bright. Please provide the wattage of the light source.

Reply: Thanks for your good comment. The fluorescent photographs of hydrogel were triggered by UV light source with photo power of 20 W in Fig. 1, Fig. 2, Fig. 3, Fig. 5, Extended Data Fig. 1, Extended Data Fig. 2, Extended Data Fig. 5, Extended Data Fig. 6, Supplementary Fig. 19, and Supplementary Fig. 20. The fluorescent photographs in Fig. 6 and Supplementary Fig. 23 were taken under a 50 W UV light source. The wavelength of the light source is 365 nm. We have added the descriptions in the “**Supplementary Text**” section of the Supplementary information.

The revision made was as follows:

“5. Light source information for taking photos

The fluorescent photographs of hydrogel were triggered by UV light source with photo power of 20 W in Fig. 1, Fig. 2, Fig. 3, Fig. 5, Extended Data Fig. 1, Extended Data Fig. 2, Extended Data Fig. 5, Extended Data Fig. 6, Supplementary Fig. 19, and Supplementary Fig. 20. The fluorescent photographs in Fig. 6 and Supplementary Fig. 23 were taken under a 50 W UV light source. The wavelength of the light source is 365 nm.”

(5) Please provide the time-resolved fluorescence decay curves of these materials to determine the nature of the luminescence.

Reply: Thanks for your good comment. As shown in **Supplementary Fig. 6**, we have supplied the time-resolved fluorescence decay curves of PNASC_{25°C} and PNASC_{100°C} hydrogels. The results showed that the fluorescence lifetimes of PNASC_{25°C} and PNASC_{100°C} hydrogels were $\tau=3.8$ ns and $\tau=17.1$ ns, respectively, which indicated that the enhanced electron delocalization ability also endows the fluorescent hydrogel with longer fluorescence life.

We have supplied a description of the fluorescence lifetime in the “**Luminescent performance of hydrogels**” section of the main manuscript. In addition, information on the corresponding instruments has been added in the “**Characterization instruments and methods**” section. The revision made was as follows:

“Besides, the fluorescence lifetime also increased from $\tau = 3.8$ ns for the PNASC_{25°C} hydrogel to $\tau = 17.1$ ns for the PNASC_{100°C} hydrogel (**Supplementary Fig. 6**).”

“The time-resolved fluorescence decay curves were recorded by means of an FLS1000 instrument (Edinburgh).”

Supplementary Fig. 6 | Time-resolved fluorescence decay curves of hydrogels. (a) The decay curves of PNASC_{25°C} and corresponding fitted data. (b) The decay curves of PNASC_{100°C} and corresponding fitted data.

(6) In Fig. 4c, there are no obvious D-A structures from the distribution of HOMO-LUMO orbits. Please provide more evidence to prove the existence of delocalization D-A.

Reply: Thanks for your good comment. The D-A structure referred to in the manuscript is the donor (D) and acceptor (A) of interchain hydrogen bonding (-N-H group for donor, -C=O group for acceptor). This is different from the traditionally electron D-A structures where the polymer backbone contains electron donor (D) and electron acceptor (A) as D-A repeating units. Therefore, there is no obvious D-A structure on the backbone of the proposed hydrogel. To avoid ambiguity, we have replaced all “D-A” with “H-bond D-A” in the revised manuscript.

In addition, the enhancement delocalization H-bond D-A of PNASC_{100°C} state is further demonstrated compared with PNASC_{25°C} state. We performed the weak interaction-IGM analysis plot (**Supplementary Fig. 18**). PNASC_{100°C} hydrogel has a clear formation of two hydrogen bonds with a blue color in the middle of the two isosurfaces. It can be seen in the scatter plot, there is a peak in the blue region corresponding to the formation of a hydrogen bond. However, PNASC_{25°C} state not only has two hydrogen bonds formed by -C=O and -NH, but also two strong isosurfaces with water molecules. The results show that the formed H-bonds between -C=O and -N-H are harassed by hydrated water molecules in the hydrogel matrix, which results in an inhibited interchain hydrogen bonding and less possibility to achieve delocalization of H-bond D-A. On the contrary, PNASC_{100°C} hydrogel possess strong H-bonding interactions to restrain the hydration of water molecules by thermodynamic promotion, with enhanced interchain delocalization for a more efficient CTE process. We added this result in **Supplementary Fig. 18**.

Supplementary Fig. 18 | Scatter graphs and isosurfaces of the noncovalent interactions of PNASC_{100%} and PNASC_{25%} by IGM method (Blue and green indicate strong and weak attraction, respectively. Red means repulsion).

Reviewer #2 (Remarks to the Author):

In this paper, Wang et al. propose a thermodynamics-driven strategy to achieve the delocalized H-bonding induced nonconventional red fluorescent emission in hydrogel. They discussed the fundamental mechanism of this emergent phenomenon in fluorescent hydrogel. Importantly, these contributions show significant implications for deeply understanding the inherent clustering-triggered emission (CTE) when forming the spatial conjugation in water surroundings. Moreover, the authors also verified the feasibility of 3D-printed red fluorescent hydrogels in applications such as fluorescence detection and soft robotics by grayscale digital light processing (g-DLP) technology. This work bridges an obvious knowledge gap and provide a new insight in understanding the inherent clustering-triggered emission (CTE) when forms the spatial conjugation in water surrounding. Therefore, it can be considered for publication in Nature Communications, but some minor comments are still required before publication.

Answer: Dear reviewer, thank you for your encouraging comments. We have revised the manuscript after carefully considering your comments and valuable suggestions. All modifications are shown in red in the revised manuscript.

1. In Fig. 2d, the PNASC_{100°C} hydrogel also exhibits a blue light emission region at approximately 425 nm, please provide additional explanation.

Reply: Thanks for your professional comment. As observed from excitation-emission contour plots images of PNASC_{25°C} and PNASC_{100°C} (Fig. 2d), PNASC_{100°C} hydrogel also exhibits a similar blue light emission region at ~425 nm by contrast with PNASC_{25°C} with a single emission center at ~490 nm. In the first place. We believe that the red shift of fluorescence originates from the heat-induced rearrangement and aggregation of polymer chains for red light emission region of PNASC_{100°C}. It can promote to build large through-space interaction by strong H-bonding interactions while restrain the hydration of water molecules (Fig. 4a). However, for blue light emission region at ~425 nm of PNASC_{100°C} hydrogel, we are of the view that not all the polymer chains can rearrangement as well as to induce tight clusters. And thus, there are also small portions

of polymer chains in the hydrogel have a similar conformation to that in PNASC_{25°C} hydrogel. Therefore, the PNASC_{100°C} hydrogel also exhibits a blue light emission region at approximately 425 nm.

Fig. 2d | Excitation-emission contour plots images of PNASC_{25°C} and PNASC_{100°C}.

Fig. 4a | Schematic diagram of the process of conformational change of PNASC hydrogels.

2. The fluorescent color depends on the temperature of post-heating. It can be found that the PNASC_{65°C} hydrogel retained blue fluorescent properties, while the PNASC_{85°C} hydrogel exhibited a sudden transition to red fluorescence. It's a large wavelength shift from blue to red. The authors should further discuss this process. And meanwhile, we are curious as to why hydrogels only show the isolated wavelength shift (i.e., blue or red) instead of a transition state for other middle wavelength.

Reply: Thanks for your good comment. We have thought deeply about this issue and investigated the literature on it. According to Tang et al.'s understanding of clusterization-triggered emission (CTE) and some relevant works, it is believed that the structure of clusteroluminogens, including interactions between groups, stacking, and size, plays a crucial role to direct fluorescence properties. For example, the luminescence wavelength of the cluster can be related to the "size" of aggregate, which

can be either the diameter of the particles or the molecular weight of polymers. It generally revealed that larger the “size” of the cluster, longer the emitted wavelength (*Materials Today*, 32, 275-292; *Advanced Materials*, 32, 2001457). However, the clusteroluminogens for this fluorescence hydrogel are gradually created under thermodynamic driven without having a transition state for other middle wavelengths, which is related to the specificity of the red fluorescence generated mechanism. The blue or red luminescence of hydrogels is governed by hydrated water molecules in the hydrogel matrix, which is not a size-dependent luminescence behavior. Therefore, the fluorescent hydrogel exhibits only two fluorescent states, i.e., red and blue fluorescence.

3. For the solvent stability of hydrogels, the paper only verifies the fluorescence stability of polymer powders. However, the fluorescence stability of hydrogel was not verified in detail. The supplementation is recommended.

Reply: Thanks for your good comment. To verify the fluorescence stability of hydrogels in different solvents, the fluorescence properties of PNASC_{100°C} hydrogel were observed after immersing it in DI water, N, N-dimethylacetamide (DMAC), N, N-dimethylformamide (DMF), and ethanol (EtOH) for 4 h, respectively. As shown in **Supplementary Fig. 20**, it can be found that hydrogel still remained stable in various solvents without losing its red fluorescent performance. However, we found that the red fluorescence of PNASC_{100°C} hydrogel gradually disappeared in DMSO solvent, which was attributed to the strong ability of DMSO to break the inter-molecular H-bonds. Finally, we deleted the corresponding DMSO part (**Fig. R1**).

Supplementary Fig. 20 | Photos of PNASC_{100°C} hydrogel in DI water, DMAC, DMF, and EtOH, respectively. Under UV light (365 nm). The scale bar is 1 cm.

Fig. R1 | Photos of PNASC_{100°C} hydrogel in DMSO. The scale bar is 1 cm.

We added this result in **Supplementary Fig. 20**, and corresponding experiment descriptions have been added to the “**Methods**” section in the revised manuscript. The revision made was as follows:

“PNASC_{100°C} hydrogels (30 mm long × 5 mm wide × 0.8~1 mm thick) were added to DI water, DMF, DMAC, or EtOH for 4 h, respectively. Then, the absorbance spectra were recorded by a Shimadzu UV-2600 spectrophotometer.”

4. In Supplementary Fig. 5, the elongation at break of the PNASC_{100°C} hydrogel decreases after heating. What’s the reason for this?

Reply: Thanks for your precious comment. The tensile strength of PNASC_{100°C} hydrogel is ~ 7.5 MPa, which is higher than that of PNASC_{25°C} hydrogel. However, the elongation at break of PNASC_{100°C} hydrogel is ~57%, which is lower than that of PNASC_{25°C} hydrogel. The aggregated clusters in hydrogel matrix contributed to the strength of the hydrogel, which is similar to the effect of nanofiller enhancement. However, the homogeneity of H-bonding network of PNASC_{100%} hydrogel s decreased owing to the aggregated clusters, which led to the limited elongation of polymer chains during stretching. So, the tensile elongation at the break of hydrogel is reduced.

5. For the Cell viability assay, the article does not mention the type of cells used, and additional clarification is needed to declare.

Reply: Thanks for your kind comment. We are very sorry for missing information about the CCK-8 experiments. We used L929 cells from Procell Life Science&Technology Co.,Ltd. for cell proliferation experiments. We have added it in the “**Methods**” section of the main manuscript.

The revision made was as follows:

“L929 cells (Procell Life Science&Technology Co.,Ltd.) were used for cell experiments.”

Reviewer #3 (Remarks to the Author):

Reply: We sincerely thank you for your time and effort to review our manuscript. We have revised all reviewers' good comments one by one to improve the quality of our manuscript. All changes and supplementary information are highlighted in red in the revised manuscript.

Reviewer #4 (Remarks to the Author):

We sincerely thank you for your time and effort to review our paper. We have revised the manuscript after carefully considering your comments and suggestions. All modifications are shown in red in the revised manuscript.

Overall the article is poorly written and requires an extensive spelling and grammar check.

Reply: Thank you for the precious comment. The article has been better edited by native speaker and the relevant certificates are attached at the end.

p. 3: No reference is made of a crosslinker. The authors refer to the use of N-acryloylsemicarbazide in the presence of an initiator to create a hydrogel. However, starting from this monomer, no hydrogel networks can be formed. Instead only linear polymer chains can be formed with lack of stability in the presence of a solvent. Furthermore, the authors should elaborate on possible side reactions occurring in DMSO as solvent when performing a radical-mediated polymerization. Furthermore, the authors should evidence hydrogel stability by referring to gel fractions obtained following polymerization. I would also recommend to analyze the amount of unreacted double bonds with HR-MAS NMR spectroscopy to complement gel fraction assays.

Reply: Thank you for your good comment. It's needed to point out that there is no crosslinker introduced in this fluorescent supramolecular hydrogel system. The light-curing hydrogel monomer is N-acryloylsemicarbazide (NASC) containing an amide and a urea group. The strong hydrogen bonding side group of NASC results in a supramolecular poly(N-acryloyl semicarbazone) (PNASC) hydrogel (**Fig. R2**). In other words, the linear polymer can be assembled into robust supramolecular networks through strong H-bonds. As shown in **Fig. R3**, the cured hydrogel (Initial-PNASC) has good solid-like character, which can endure the compressed or cut without obvious deformation and solvent spillage. In addition, after immersing in water, H-bonding of

hydrogel is further enhanced by phase conversion to generate superior mechanical properties. Such supramolecular PNASC hydrogel systems have been previously studied in detail by Liu et al. and our group but all reported works didn't use any crosslinkers (*Mater. Horiz.*, 2020, 7, 1160-1170; *Adv. Funct. Mater.* 2023, 33, 2210395). On the contrary, we also found that the mechanical properties of water equilibrium PNASC hydrogels with crosslinker (e.g. BIS or PEGDA) were dramatically reduced. We hypothesized that the reconfiguration of the H-bond by supramolecular self-assembly process is restrained by a 3D cross-linker network, which hinders the proximity of polymer chains and the formation of hydrogen bonding structures. In short, based on the concept of supramolecular chemistry, we didn't introduce any crosslinker in the proposed supramolecular fluorescent hydrogel.

Fig. R2 | Schematic diagram of the gelation process.

Fig. R3 | (a) Digital photographs of initial-PNASC hydrogel. Digital photographs of hydrogel (b) compression and (c) cut process. The scale bar is 1 cm.

The NASC monomers were dissolved in different content of mixed solvents (DMSO/DI

water) with 0 wt%, 30 wt%, 50 wt%, 70 wt% and 100 wt% of DMSO solvent, and the obtained water equilibrium hydrogels were named as-PNASC_{DMSO-0%}, as-PNASC_{DMSO-30%}, as-PNASC_{DMSO-50%}, as-PNASC_{DMSO-70%}, and as-PNASC_{DMSO-100%}. As shown in **Fig. R4**, Fourier transform infrared (FT-IR) characterizations confirmed that there were no differences in the structural composition of polymers except for a slight variation in the strength of the characteristic peaks. It was maybe affected by interchain hydrogen bonding due to different solvents. The characteristic peaks of as: 3340 (NH), 3186 (NH), 3033 (NH).

In general, we believed that the conversion rate of monomer polymerization is related to the time of polymerization, longer polymerization time, less monomer remains. For this supramolecular PNASC hydrogel, the final obtained hydrogel was stable after water equilibrium. If there is any monomer residue, it was also lost through solvent exchange. It can also be seen from **Fig. R4** that there is no obvious monomer residue (i.e., -C=C at 1625 cm⁻¹). Furthermore, the convincing evidence from previous studies also confirmed the stability of hydrogel in structure or property aspects after water equilibration (*Mater. Horiz.*, 2020, 7, 1160-1170; *Adv. Funct. Mater.* 2023, 33, 2210395).

Fig. R4 | FTIR spectra of as-PNASC_{DMSO-0%}, as-PNASC_{DMSO-30%}, as-PNASC_{DMSO-50%}, as-PNASC_{DMSO-70%}, and as-PNASC_{DMSO-100%}.

Materials & Methods

p.17: What yield was obtained for the monomer synthesis?

Reply: Thank you for the good comment. We have calculated the yield of monomer

synthesis, and the monomer average yield was ~66%. We have added it in the methods section.

The revision made was as follows:

“The average yield of NASC was ~66%.”

p.20: How was DLP optimized? What was the resolution/CAD-CAM mimicry observed? It is surprising that the authors did not require photo-absorber for DLP (since this is typically applied in DLP to ensure appropriate resolution).

Reply: Thank you for the good comment. Admittedly, photo-absorber is an important component for DLP to achieve high printing resolution. However, this mixed solvent system has better print accuracy and stability (less prone to dehydration) compared to pure water systems as common. We have optimized the printing parameters, including light source intensity, exposure time, thickness of each layer, and initiator content for DLP printing. The printed hydrogel structures had satisfactory resolution based on our homemade DLP printing system. It is enough for us to perform the application paradigms of hydrogel. In this work, we focus on developing a novel red fluorescent hydrogel and understanding the CTE mechanism instead of manufacturing hydrogel machines. Of course, we will further demonstrate the additive manufacturing of this proposed hydrogel, which may be important for fabricating hydrogel components and devices.

In particular, we have found that the content of LAP initiator content is of critical importance for printing resolution and is preferably controlled within 0.3 wt% content of the monomer mass. However, for hydrogel printing with complex and sophisticated structures, a small amount of tartrazine was added into printing ink as a photo-absorber only for the hydrogel castle and Eiffel tower to improve the printing accuracy (**Fig. 5c**). The LAP is a typical blue light initiator triggered by 405 nm light. We added photo-absorber tartrazine owing to the compatible and matched absorption wavelength at 400 nm. The wavelength of photo-absorber didn't overlap with the emission wavelength of hydrogel. Therefore, if someone want to improve the printing resolution of hydrogel for manufacturing complex and fine structures, it is allowed to add some photo-absorber

in this hydrogel ink for DLP 3D printing.

To confirm the abovementioned fact, the printed structured hydrogels at different scales further illustrate the printing fidelity (**Fig. R5**). As shown in **Fig. R5a, b,** and **c**, the hydrogel ink, both with and without photo-absorber, has excellent printability. Specifically, we designed a series of cubic models with features of 1 mm, 0.75 mm, 0.5 mm, and 0.25 mm to demonstrate the resolution/CAD-CAM mimicry. It can be found that the sample printed by ink absenting photo-absorber showed dimensions of 1.08 ± 0.02 mm, 0.85 ± 0.02 mm, 0.60 ± 0.02 mm, and 0.45 ± 0.01 mm while the structure printed by ink containing photo-absorber exhibited the feature of for the while 0.92 ± 0.03 mm, 0.76 ± 0.01 mm, 0.58 ± 0.02 mm and 0.37 ± 0.01 mm (**Fig. R6**). The difference is about 17% for both hydrogel inks. The results show that both printing ink have very good printability and printing resolution. It's needed to be pointed out that photo-absorber is necessary for high-resolution 3D printing.

Fig. R5 | (a) Test model for printing. The printed cubic structure models of 1 mm, 0.75 mm, 0.5 mm, and 0.25 mm dimensions. (b) Digital photographs of printed hydrogels with photo-absorber and not. (c) Photographs of different scale structures of printed hydrogels under the optical microscope.

Fig. R6 | Size comparison of hydrogels printed with photo-absorber and not.

We have supplied the specific printing conditions in the “**Supplementary Text**” section of the Supplementary information. The revision made was as follows:

“4. Structured fluorescence hydrogel printing

The hydrogel structure was fabricated by DLP 3D printing using hydrogel ink consisted of monomer NASC and photo-initiator LAP. The monomer NASC were dissolved in the mixed solvent of DMSO and DI (7/3, wt/wt), then LAP (0.3 wt% of the monomers) was added to mix by magnetic stirring at nitrogen atmosphere. The hydrogel structure was manufactured by a homemade DLP 3D printer with a 405 nm DMD. The curing time of each layer (5 s), light source (2K resolution, 1 W power) and layer thickness (0.1 mm) remained constant for all experiments. The as-prepared structured hydrogels were thoroughly immersed in DI water at 100 °C for 48 h. In this case, to further improve print accuracy, a small amount of photo-absorber (0.25 g/L tartrazine) was added to the above hydrogel ink for printing of hydrogel castle and Eiffel tower structure (**Fig. 5c**).”

The materials & methods section reports on cell viability assays, but no results are mentioned in the article body regarding the execution of such experiments.

Reply: Thank you for your good comment. The description of the experimental results is in the “**Red fluorescent hydrogel structures and responsiveness**” section. The

results are “In addition, Cell Counting Kit-8 (CCK-8) cell viability tests were also conducted to test the biocompatibility of the PNASC_{100°C} hydrogel. As shown in **Supplementary Fig. 21**, the proposed fluorescent hydrogels maintained a cell viability of $95.30 \pm 2.93\%$ after 7 days, demonstrating favorable cell biocompatibility.”

Furthermore, reviewer 2 raised questions about experimental cell types. We have added it in the “**Methods**” section of the main manuscript. The revision made was as follows:

“L929 cells (Procell Life Science&Technology Co.,Ltd.) were used for cell experiments.”

REVIEWER COMMENTS

Reviewer #1 (Remarks to the Author):

1. I have reviewed the authors' responses to reviewer 4, and it appears that they have adequately addressed the questions raised by this reviewer.
2. I appreciate the authors' efforts in addressing most of the concerns I raised (reviewer 1). However, I do not completely agree with the explanation the authors provided for the luminescence mechanism (for the second issue). I kindly suggest the authors meticulously revise the description of the luminescent mechanism in this work before publication.

While acknowledging the significance of hydrogen bonding in achieving excellent CTE, I strongly suggest that the authors avoid solely attributing their luminescence mechanism to the electron delocalization of interchain strong multiple H-bonding interactions. If the delocalization of electrons through hydrogen bonding is indeed efficient, it raises the question of why the luminescence quantum efficiency is so low in this hydrogen-bond-rich system and, meanwhile, why the ice is not emissive (who shows one of the strongest H-bond). It is evident that a substantial number of heteroatoms participate in hydrogen bond formation in this system, creating a conducive environment for clusteroluminescence. However, it is worth noting that a small fraction of heteroatoms that have not formed hydrogen bonds may potentially engage in n-n/ π TSI. The contribution of this aspect should not be overlooked, particularly in a weak clusteroluminescence system.

In relation to the experiments described by the authors (i.e., adding DMSO), it is important to acknowledge that the breaking of hydrogen bonds will simultaneously disrupt other weak interactions (such as n-n TSI) due to the more flexible environment. Meanwhile, EtOH has a stronger ability to destroy H-bond than DMSO, but red luminescence is still observed in EtOH. Consequently, these experiments may not offer convincing evidence to support the dominant role of the H-bond in electron delocalization.

As a result, in the main text, it is preferable for the authors to attribute the nonconventional red fluorescent emission to contributions of various heteroatom-involved interactions (including heteroatom-involved hydrogen bonds and n-n/ π TSI), rather than qualitatively attributing it to a specific singular H-bonding contribution.

Reviewer #2 (Remarks to the Author):

I have read this revised manuscript very carefully, feeling the manuscript of the current version is well prepared for publication.

Reviewer #3 (Remarks to the Author):

Response to Reviewers

Reviewer #1 (Remarks to the Author):

1. I have reviewed the authors' responses to reviewer 4, and it appears that they have adequately addressed the questions raised by this reviewer.

2. I appreciate the authors' efforts in addressing most of the concerns I raised (reviewer 1). However, I do not completely agree with the explanation the authors provided for the luminescence mechanism (for the second issue). I kindly suggest the authors meticulously revise the description of the luminescent mechanism in this work before publication.

While acknowledging the significance of hydrogen bonding in achieving excellent CTE, I strongly suggest that the authors avoid solely attributing their luminescence mechanism to the electron delocalization of interchain strong multiple H-bonding interactions. If the delocalization of electrons through hydrogen bonding is indeed efficient, it raises the question of why the luminescence quantum efficiency is so low in this hydrogen-bond-rich system and, meanwhile, why the ice is not emissive (who shows one of the strongest H-bond). It is evident that a substantial number of heteroatoms participate in hydrogen bond formation in this system, creating a conducive environment for clusteroluminescence. However, it is worth noting that a small fraction of heteroatoms that have not formed hydrogen bonds may potentially engage in n-n/ π TSI. The contribution of this aspect should not be overlooked, particularly in a weak clusteroluminescence system.

In relation to the experiments described by the authors (i.e., adding DMSO), it is important to acknowledge that the breaking of hydrogen bonds will simultaneously disrupt other weak interactions (such as n-n TSI) due to the more flexible environment. Meanwhile, EtOH has a stronger ability to destroy H-bond than DMSO, but red luminescence is still observed in EtOH. Consequently, these experiments may not offer convincing evidence to support the dominant role of the H-bond in electron delocalization.

As a result, in the main text, it is preferable for the authors to attribute the nonconventional red fluorescent emission to contributions of various heteroatom-involved interactions (including heteroatom-involved hydrogen bonds and n-n/ π TSI), rather than qualitatively attributing it to a specific singular H-bonding contribution.

Reply: We sincerely thank you for your time and efforts in reviewing our paper. We have revised the manuscript carefully based on your good comments and valuable suggestions. All modifications are shown in red in the revised manuscript.

Thanks for your professional comments. As shown in **Fig. R1**, according to as-PNASC_{DMSO-0%} hydrogels with severe phased-separation structures aggregated by intrachain multiple H-bonding (**Supplementary Fig. 10**). Its tight clusters contained many n or π interactions in the -C=O and -N-H moieties. However, there was no obvious red fluorescence but only a small amount of red fluorescence. We believe that the contribution of interchain hydrogen bonding is a prerequisite for the formation of red fluorescence.

At the same time, we strongly agree with reviewer 1's viewpoint about the contribution of n-n, n- π interactions of TSI toward red fluorescence. Based on previous theoretical on CTE, we also agree on the contribution of heteroatoms to cluster luminescence systems. In our system, it is also worth noting that a substantial number of heteroatoms (O and N) participate in hydrogen bond formation in the hydrogel, which also plays a critical role in building a favorable environment for cluster luminescence. Besides, a small fraction of heteroatoms containing lone-pair electrons that do not form strong H-bonds may also participate in n-n, n- π interactions of TSI. So, the contribution of this part should not be overlooked (**Supplementary Fig. 19**). Therefore, the proposed unconventional red fluorescence in the hydrogel is attributed to various heteroatom-involved interactions (including heteroatom-involved hydrogen bonds and n-n/ π TSI) instead of qualitatively attributing it to a specific singular H-bonding contribution.

Fig. R1 | Digital photos of as-PNASC_{DMSO-0%} and PNASC_{DMSO-0%} hydrogels under

natural light and UV light (365 nm). The as-PNASC_{DMSO-0%} was heated at 100°C for 48 h and named PNASC_{DMSO-0%}. The scale bar is 5 mm.

Supplementary Fig. 19 | Interaction of cluster luminescence in PNASC_{red} hydrogels.

We added this result in **Supplementary Fig. 19** and the revised description of the luminescent mechanism has been added in the revised manuscript. The revision made was as follows:

In the “**Main**” section

“Besides, a small fraction of heteroatoms (O and N) containing lone-pair electrons that do not form strong H-bonds may also participate in *n-n*, *n-π* interactions of TSI.”

In the “**Luminescence mechanism of the hydrogels**” section

“In addition, it is also worth noting that a substantial number of heteroatoms participate in hydrogen bond formation in the hydrogel, which also plays a critical role in building a favorable environment for cluster luminescence. Besides, a small fraction of heteroatoms containing lone-pair electrons that do not form strong H-bonds may also participate in *n-n*, *n-π* interactions of TSI. So, the contribution of this part should not be overlooked (**Supplementary Fig. 19**)”

In the “**Conclusion**” section

“The electron-rich groups in the proposed hydrogel extended the electron delocalization through intermolecular charge transfer to achieve stable red fluorescence emission in a water environment by various heteroatom-involved interactions (including heteroatom-involved hydrogen bonds and *n-n/π* TSI).”

REVIEWERS' COMMENTS

Reviewer #1 (Remarks to the Author):

I am satisfied with the revised version and gladly recommend it to be published in Nature Communications. However, I still have a suggestion about the title. I suppose that it is insufficient to only mention the importance of hydrogen bonds in the title. There is no direct evidence that the delocalization of H-bonding contributes much to this nonconventional luminescence. To avoid academic controversy, I kindly suggested that it is better to change the title "Delocalized H-bonding induced nonconventional red fluorescence emission in hydrogels" to " Multiple hydrogen-bonding induced nonconventional red fluorescence emission in hydrogels".

Reviewer #3 (Remarks to the Author):

Response to Reviewers

Reviewer #1 (Remarks to the Author):

Reply: We highly appreciate the reviewer #1's positive comments and helpful suggestions. We have modified the title to "Multiple hydrogen-bonding induced nonconventional red fluorescence emission in hydrogels". The suggested title could more clearly and comprehensively summarize the fundamental science in this paper.

Reviewer #3 (Remarks to the Author):

Reply: We sincerely thank you for your time and efforts in reviewing our paper.